Optimizing transformer-based prediction of human microbe–disease associations through integrated loss strategies

http://orcid.org/0000-0002-0687-7343 Zhu Rong 1 zhurongsd@qfnu.edu.cn
Wang Yong 2
Shang Junliang 1
Dai Ling-Yun 1
Li Feng 1
1 School of Computer Science, Qufu Normal University , Rizhao, Shandong , China
2 Laboratory Experimental Teaching and Equipment Management Center, Qufu Normal University , Rizhao, Shandong , China
Wan Shibiao
Electronic publication date: 2025 Aug 15
Publication date: 2025
Volume: 11
Electronic Location ID: e3098
Received 2024 Nov 18; Accepted 2025 Jul 11
Copyright: © 2025 Zhu et al.
Copyright year: 2025
Copyright holder: Zhu et al.
License: This is an open access article distributed under the terms of the Creative Commons Attribution License, which permits unrestricted use, distribution, reproduction and adaptation in any medium and for any purpose provided that it is properly attributed. For attribution, the original author(s), title, publication source (PeerJ Computer Science) and either DOI or URL of the article must be cited.
License URL: https://creativecommons.org/licenses/by/4.0/

Keywords: HGNNs, Transformer, Huber loss, Contrastive learning, Microbe-disease association prediction

Funding: Shandong Social Science Planning 21BTQJ02 This work was funded by the Shandong Social Science Planning Fund Program (No. 21BTQJ02). The funders had no role in study design, data collection and analysis, decision to publish, or preparation of the manuscript.

==============================
Microorganisms play an important role in many complex diseases, influencing their onset, progression, and potential treatment outcomes. Exploring the associations between microbes and human diseases can deepen our understanding of disease mechanisms and assist in improving diagnosis and therapy. However, traditional biological experiments used to uncover such relationships often demand substantial time and resources. In response to these limitations, computational methods have gained traction as more practical tools for predicting microbe-disease associations. Despite their growing use, many of these models still face challenges in terms of accuracy, stability, and adaptability to noisy or sparse data. To overcome the aforementioned limitations, we propose a novel predictive framework, HyperGraph Neural Network with Transformer for Microbe-Disease Associations (HGNNTMDA), designed to infer potential associations between human microbes and diseases. The framework begins by integrating microbe–disease association data with similarity-based features to construct node representations. Two graph construction strategies are employed: a K-nearest neighbor (KNN)-based adjacency matrix to build a standard graph, and a K-means clustering approach that groups similar nodes into clusters, which serve as hyperedges to define the incidence matrix of a hypergraph. Separate hypergraph neural networks (HGNNs) are then applied to microbe and disease graphs to extract structured node-level features. An attention mechanism (AM) is subsequently introduced to emphasize informative signals, followed by a Transformer module to capture contextual dependencies and enhance global feature representation. A fully connected layer then projects these features into a unified space, where association scores between microbes and diseases are computed. For model optimization, we propose a hybrid loss strategy combining contrastive loss and Huber loss. The contrastive loss aids in learning discriminative embeddings, while the Huber loss enhances robustness against outliers and improves predictive stability. The effectiveness of HGNNTMDA is validated on two benchmark datasets—HMDAD and Disbiome—using five-fold cross-validation (5CV). Our model achieves an AUC of 0.9976 on HMDAD and 0.9423 on Disbiome, outperforming six existing state-of-the-art methods. Further case studies confirm its practical value in discovering novel microbe–disease associations.

Introduction

Microorganisms, including bacteria, fungi, archaea, and viruses, play significant roles in various biological processes despite their small size and simple structure (de Vos et al., 2022). Recent research has revealed that the vast number of microorganisms within the human body has a profound impact on health. These microbial communities and their genes, collectively known as the microbiome, are present throughout the body and are critical to human health and disease (The Human Microbiome Project Consortium, 2012).

Understanding the relationships between microorganisms and diseases is vital for uncovering disease mechanisms and developing new therapies. Traditional biological experimental methods, although invaluable, are often time-consuming and labor-intensive, making it challenging to process large datasets efficiently. In response to ongoing challenges in understanding microbe-disease relationships, recent years have witnessed a surge in the use of artificial intelligence to uncover hidden biological associations.

Chen et al. (2017) introduce the KATZHMDA model. They designed a heterogeneous network using KATZ path scoring and enhanced it with Gaussian Interaction Profile (GIP) kernel similarity metrics. A global random walk strategy was applied across this network. The model showed promising predictive abilities. However, the inability to generalize toward unknown nodes weakened its broader applicability. Yin et al. (2022) proposed the Network Consistency Projection and Label Propagation (NCPLP) framework. They aimed to improve predictive resilience by integrating data from Medical Subject Headings (MeSH) and 16S rRNA gene sequences. This was achieved through a combination of network consistency projection and label propagation algorithms. Although this approach enhanced the model’s robustness under sparse data conditions, its reliance on the MeSH system posed a bottleneck. Liu et al. (2023) introduced a fresh perspective by proposing Multi-Similarity Information Fusion Through Low-Rank Representation to Predict Disease-Associated Microbes (MSF-LRR). Unlike prior methods, MSF-LRR synergizes diverse similarity measures through the use of low-rank representation (LRR). Although effective at extracting local structural patterns, the model’s linear fusion strategy hindered its ability to capture nuanced distinctions among similarity sources. The Multiple Similarities and LINE Algorithm (MSLINE) method (Wang et al., 2022) integrates Large-scale Information Network Embedding (LINE) embeddings and multiple similarity metrics to build a heterogeneous association network, successfully extracting structural features, though it struggles with generalizability to novel diseases. Meanwhile, Heterogeneous Network and Global Graph Feature Learning (HNGFL) (Wang, Lei & Pan, 2022) employs GraRep embeddings and support vector machines to mine high-dimensional features, excelling at global feature learning but remaining vulnerable to sparse data distributions. BPNNHMDA (Li et al., 2021) introduces a backpropagation neural network (BPNN) architecture, where GIP-derived similarity is used to initialize weights and enhance convergence. Although it is among the first neural models in this domain, its fixed learning rate and predetermined initialization strategy may reduce adaptability. Prediction of Microbe-Disease Associations Based on Deep Matrix Factorization Using Bayesian Personalized Ranking (DMFMDA) (Liu et al., 2021) combines neural network embeddings with Bayesian personalized ranking (BPR), blending the memorization strength of matrix factorization with the generalization capacity of deep learning. This makes it well-suited for ranking tasks in sparse association settings. Multi-View Feature Aggregation for Predicting Microbe-Disease Association (MVFA) (Peng et al., 2023) implements a multi-view learning approach by integrating non-negative matrix tri-factorization, dual random walks, and capsule networks to capture both linear and non-linear relational patterns. Despite improved performance, its strategy for aggregating multi-view features lacks fine-grained control. WMGHMDA (Long & Luo, 2019) introduces weighted meta-graphs to model propagation pathways in heterogeneous information networks, offering nuanced edge-weighting. However, its complexity and reliance on detailed network design may limit its scalability. NTSHMDA (Luo & Long, 2020) applies network topological similarity and an improved random walk method to address class imbalance. Yet, the model is constrained by its dependence on known topological structures, which hampers prediction for isolated nodes. Yan et al. (2020) introduced BRWMDA, which integrates bi-random walk processes with symptom-based data and a similarity adjustment mechanism grounded in a logistic function. Although the integration of Gaussian Interaction Profile (GIP)-based and phenotype-related data enhances the overall utility of predictive models, but it cannot predict associations for microbes that have not been documented before. Huang et al. (2017) introduced the NGRHMDA model. By combining neighborhood collaborative filtering with graph-centric scoring methods, it yields improved predictions for frequently studied microbes. But it tends to overfit due to its dependence on high-frequency associations, thereby compromising its adaptability to novel instances. In 2021, Long et al. (2021) introduced GATMDA, a method that leverages the representational power of graph attention networks (GAT) combined with inductive matrix completion. This approach enables the model to handle novel microbial and disease instances more effectively. Yet, its performance may suffer when input data suffer from sparsity or missing biological descriptors. In 2023, Wang et al. (2023) proposed Graph Convolutional Neural Network with Multi-Layer Attention (GCNMA) Mechanism for Predicting Potential Microbe-Disease Associations. It employs a graph convolutional architecture enriched by multilayer attention schemes. It has shown impressive predictive strength across multiple datasets. But due to its performance is highly dependent on the structural integrity of the input graph, when any distortion or incompleteness in the graph can significantly impair performance.

Over the last 10 years, the field of computational biology has witnessed substantial advancements in techniques aimed at identifying links between microbial species and human diseases. Yet, these methods are still not perfect. Such as sparse data matrices continue to hinder robust model training, similarity-based learning often suffers from imbalance in input distributions, and biases introduced by skewed similarity distributions can further complicate matters.

To address the limitations inherent in prior methods, this study proposes a new method called HyperGraph Neural Network with Transformer for Microbe-Disease Associations (HGNNTMDA). The method employs hypergraph neural networks (HGNNs) that incorporate Transformer and contrastive learning. The method employs Huber loss as the training objective. The primary contributions of this article are: The proposed framework integrates both K-nearest neighbor (KNN) graphs and K-means clustering-derived hypergraphs. This combination ensures that both fine-grained and holistic topological information is captured effectively.

The approach fuses hypergraph convolutional operations with contrastive learning. The former enables the model to interpret multifaceted relationships that extend beyond pairwise links, while the latter promotes consistency across different graph views by optimizing the similarity of learned embedding. This synergy boosts feature discrimination across complex biological entities.

The approach is modeled through a Transformer encoder. Using self-attention, this component allows the model to evaluate interdependencies across distant nodes, ultimately improving the contextual expressiveness of the learned representations.

The Huber loss function was employed due to its ability to stabilize gradient descent, thereby enhancing training reliability. Its resilience to noise and insensitivity to statistical outliers make it particularly suitable for robust predictive modeling.

Materials and Methods

Dataset

To construct the experimental foundation of this study, we relied on two extensively validated microbial association databases that are frequently cited in biomedical informatics research. Specifically, the Human Microbe–Disease Association Database (HMDAD) dataset curated by Ma et al. (2017), accessible at http://www.cuilab.cn/hmdad, and the Disbiome resource introduced by Janssens et al. (2018), available via https://disbiome.ugent.be/home, were selected due to their broad coverage of known microbe-disease relationships and structured annotation formats. It is worth noting that both datasets are publicly accessible and have been widely adopted in related studies, providing a reliable benchmark for evaluating predictive models in this domain.

The HMDAD database has meticulously gathered microbe–disease association data from published literature. At the time of its initial compilation, the dataset comprised a total of 483 verified interactions, representing 292 distinct microbial taxa and 39 categories of human diseases. After undergoing preprocessing and normalization steps, 450 associations were ultimately retained for use in this study. Annotated with standardized biomedical terminologies such as MeSH, HMDAD serves as a valuable resource for studying microbial involvement in human pathologies.

Disbiome, developed by Janssens et al. (2018), centers on the concept of microbial dysbiosis in various disease states. It aggregates a total of 5,573 associations from scientific publications. Following deduplication and data cleaning, 4,351 associations were preserved, covering 1,052 microbial taxa and 218 disease categories. Compared to HMDAD, Disbiome offers broader coverage and further distinguishes itself by providing information on the direction of microbial abundance changes—indicating whether a microbe is enriched or depleted in a given disease context—which enhances its utility in dysbiosis-related research.

Since neither database provides feature vectors directly, researchers typically need to construct their representations using supplementary biological information or computational techniques. In this study, we adopted the following feature processing approach. Microbial features were initially extracted from a text file to form a static attribute matrix, which may include taxonomic, functional, or genomic information. Similarly, disease features were loaded from a separate file and may reflect MeSH codes, phenotypic descriptors, or gene-disease associations. To augment the expressiveness of these features, we further applied Gaussian kernel-based similarity modeling. Specifically, Gaussian similarities were computed between microbes and between diseases based on the microbe-disease interaction matrix. These similarity matrices were then integrated with the original features to generate updated microbe and disease similarity representations. This fusion strategy preserves the original attribute information while incorporating structural insights derived from relational data, thereby improving the model’s capacity to capture potential microbe-disease associations. By combining static features with network-based structural similarity, this method provides high-quality feature vectors for downstream modeling tasks.

HGNNTMDA model overall structure

The HGNNTMDA model undertakes feature extraction and association prediction for microbial and disease data by integrating various deep learning modules, including hypergraph convolutional neural network (HGCN), attention mechanism (AM), and Transformer. The process involves several critical steps: (1) constructing the graph adjacency matrix using KNN and K-means methods to generate the joint hypergraph; (2) extracting microbial and disease features with the contrastive learning hypergraph convolutional network (CL_HGCN) and calculating the contrastive learning loss; (3) enhancing feature representation through the AM; (4) encoding and processing the attention-enhanced features using the Transformer encoder; (5) mapping these refined features into the final embedding space via a fully connected layer; and (6) calculating the similarity scores between microbial and disease features. Figure 1 illustrates overall of the HGNNTMDA.

Figure 1 Diagram of the HGNNTMDA model.

Constructing hypergraph

A hypergraph (Berge, 1984), as introduced by Berge, extends conventional graph theory by permitting each hyperedge to simultaneously link more than two nodes. This structural flexibility enables more accurate representation of complex and higher-order relationships. In our framework, we construct graph representations using two complementary strategies. The first employs the KNN (Guerraoui et al., 2023) algorithm to capture local geometric structure, where each node is linked to its k closest neighbors based on a distance metric—typically Euclidean distance—resulting in a sparse adjacency matrix that encodes proximity-based relationships. The second strategy applies K-means clustering to reveal global structural patterns by grouping nodes with similar features into clusters, each of which is treated as a hyperedge. This yields a hypergraph incidence matrix, in which a single hyperedge can simultaneously connect multiple nodes within the same cluster. Together, these approaches enable the model to jointly capture both local continuity and higher-order semantic associations in the data.

The adjacency matrix of the KNN graph can be constructed:

(1) AijKNN={1,ifxj∈Nk(xi)orxi∈Nk(xj)0,otherwise

where Nk(xi) represents the set of K nearest neighbors of node xi in the feature space.

Construct the hypergraph correlation matrix as follows:

(2) Hik={1,ifvi∈Ck0,otherwise

where vi represents the i-th node, Ck refers to the k-th cluster.

Design CL_HGCN

HGCN (Gao et al., 2023) is a specific implementation of HGNNs, focusing on feature learning through convolutional operations on hypergraph structures to capture higher-order relationships. In the proposed HGNNTMDA framework, we integrate hypergraph convolutional mechanisms with contrastive learning strategies. This combination enables the model to capture informative structural features from hypergraph data while simultaneously enhancing the quality of learned representations through contrastive optimization.

Based on the hypergraph incidence matrix H constructed in the preceding section, the normalized hypergraph Laplacian matrix L is calculated as follows:

(3) L=Dv−1/2HWDe−1HTDv−1/2

where Dv represents the degree matrix of the node, De represents the degree matrix of the hyperedge, and W denotes the hyperedge weights matrix.

Given the input node features matrix X, the convolution operation on the hypergraph can be expressed as:

(4) X′=ReLU(LXWθ)

where Wθ denotes a learnable weight matrix, ReLU is the activation function. X′ is the output feature representation after one hypergraph convolution layer.

In recent years, contrastive learning (Le-Khac, Healy & Smeaton, 2020), as an emerging learning method, has attracted great interest from a wide range of researchers. Contrastive learning is a discriminative representation learning framework or method based on the idea of contrast, which can be regarded as a self-supervised learning method aiming to learn good feature representations from unlabeled data. Unlike traditional supervised learning, contrastive learning does not require explicitly labeled labels, but instead performs self-supervised learning by exploiting the structure and features of the data itself. The basic idea of contrastive learning is to maximize the similarity between similar samples and minimize the similarity between dissimilar samples as a way to learn a more robust and generalized feature representation. Contrastive learning (CL) does not rely on the labeling of tokens and learns representations of positive and negative samples from data. It aims to bring similar samples closer to each other, making the positive and negative samples farther apart through training, thus improving the classification. As the most common information carrier in real scenarios, graph data contains rich information, so the analysis and research of graph data have important value. Contrastive learning methods have demonstrated efficient and stable performance on graph data processing.

Our framework integrates SimCLR (Chen et al., 2020). By comparing the original input with the input that has been augmented with data, SimCLR is able to better capture the intrinsic changes in the data and learn a more robust feature representation. Bidirectional similarity computation is applied to both positive and negative sample pairs. By pulling similar samples closer and pushing dissimilar samples apart, the model learns discriminative and compact embedding representations.

First, the normalized cosine similarity is used to compute the similarity between two embedded representations:

(5) sim(zi,zj)=zi⋅zj||zi||||zi||

where zi and zj represent feature representations embedded in the space. ⋅ represents the dot product of a vector. ‖‖ represents the corresponding Euclidean norm (i.e., their L2 norm).

Assuming there are 2N samples in the batch, the loss formula for each positive sample pair (zi,zj) can be expressed as:

(6) CL_loss=−logexp⁡(sim(zi,zj)/C)∑k=12Nexp⁡(sim(zi,zk)/C)

where C is a temperature parameter, usually taken as a small positive number (such as 0.1 or 0.2), which controls the smoothness of the similarity distribution. A smaller C will make the distinction between similar sample pairs more distinct. k represents other samples in the batch (including negative samples).

Attention mechanism

The AM (Brauwers & Frasincar, 2023) is designed to better capture important information and structure in the input data. AM is not a separate and complete model, but a technique that can be embedded into any model. By using this technique, the model can be targeted to learn the important information and thus improve the overall performance of the model. When dealing with huge input data, the computing speed of the neural network will be significantly reduced, in order to improve the training speed, the AM can be introduced, the core idea of the mechanism is to select the most critical part of the information to be processed, so as to accurately capture the most important features in the information, and realize the improvement of the computational efficiency. The AM is essentially a weighted processing. Integrating the AM module into HGCN can significantly enhance in the model’s capacity to allocate suitable weights to key nodes and edges. The AM allows the model to focus more precisely on the most relevant features, leading to better accuracy and efficiency.

In our proposed framework, attention is integrated at the channel level. Specifically, two separate streams of node embeddings, which are derived from distinct graph learning branches, as independent feature channels. These channels are first concatenated to form a joint representation. Following this, global average pooling is performed on each to extract high-level descriptors that reflect their respective activation patterns. These global features are subsequently fed into a two-layer feedforward neural network. The first layer applies a nonlinear transformation that helps uncover deeper semantic cues, while the second layer employs an activation function (e.g., sigmoid) to generate normalized importance scores. The final output is a recalibrated feature representation refined by the AM, which improves the model’s ability to capture meaningful patterns for downstream tasks such as representation learning or classification.

We concatenate the two feature matrices along the channel dimension, then transpose the result and denote it as X. Subsequently, we perform global average pooling on each channel of X to obtain channel-level statistics, which is formally defined as follows:

(7) zc=1d⋅n∑i=1d∑j=1nX1,c,i,jforc=1,2

where zc represents the global average of the c-th channel. X1,c,i,j represents the value at positions i and j of channel c. d represents the feature dimension. n represents the number of samples.

Subsequently, the relative importance of each channel is captured using a two-layer fully connected neural network. The first layer performs a linear transformation to generate intermediate hidden representations. The calculation is as follows:

(8) Hiddenlayer_output=ReLU(W1⋅z+b1)

where W1 and b1 represent the parameters of the first layer linear transformation.

To determine the final attention weights assigned to each feature channel, a secondary linear transformation is applied. The calculation is as follows:

(9) s=Sigmoid(W2⋅h+b2)

where W2 and b2 represent the parameters of the second layer linear transformation. The Sigmoid compresses continuous values into the interval (0, 1), ensuring bounded importance scores that can be interpreted as probability-like weights.

Once calculated, these attention weights are used to rescale the original feature representations on a channel-wise basis. A new weighted feature map is generated through this process, with each channel being adjusted based on its learned significance. The computation is as follows:

(10) X^c=sc⊗Xcforc=1,2

where sc represents the attention weight of the c-th channel (ranging between 0 and 1). X^c represents the weighted feature map of the c-th channel. ⊗ denotes channel-wise multiplication.

After obtaining the weighted features, an essential subsequent step is to transpose and concatenate them. These processed features are then fed into a Transformer encoder for more in-depth semantic analysis. As Xu, Zhu & Clifton (2023) highlighted in 2023, the Transformer encoder, built upon the multi-head self-attention architecture, has the unique ability to examine multiple aspects of the input simultaneously. It focuses on different representational subspaces, allowing it to uncover complex dependencies among features that might be missed by other methods.

Loss function selection

In the training of deep neural networks, selecting an appropriate loss function plays a pivotal role, as it significantly influences convergence rate, model fitting capacity, and generalization ability. One widely adopted loss metric is the mean absolute error (MAE), also referred to as the L1 loss, which computes the average of the absolute discrepancies between predicted outcomes and actual target values. A lower L1 loss indicates improved alignment between the model predictions and ground truth data. Due to its constant gradient, L1 loss contributes to stable training dynamics and is inherently robust against anomalous data points. However, this stability often comes at the cost of slower convergence. The formal mathematical formulation of L1 loss is given as follows:

(11) L1_loss=1n∑i=1n|yi−y^i|

where yi represents the true value, y^i is the predicted value, and n is the number of samples.

Mean squared error (MSE), also known as L2 loss function. It quantifies the discrepancy between the predicted output and the actual value by averaging the squared differences across all observations. The formal expression of L2 loss is defined as:

(12) L2_loss=1n∑i=1n(yi−y^i)2.

Huber loss combines the advantages of L2 Loss and L1 Loss. The core idea of Huber loss is to penalize smaller errors with squared error and switch to absolute error for larger errors. This design makes Huber loss not be overly affected by squared error when facing outliers, and also avoids the problem of slower convergence of absolute error. It introduces a threshold parameter, when the error is less than the threshold, L2 Loss is used, and when the error is greater than or equal to the threshold, the L1 Loss is used. By adjusting the threshold parameter, Huber loss can balance between L2 Loss and L1 Loss, so as to adapt to different data distributions and modeling needs. The definition of Huber loss is as follows:

(13) Huber_loss(a)={12a2if|a|≤δδ(|a|−12δ)otherwise

where a is the difference between the predicted and actual values, and δ is the threshold parameter.

To improve its ability to infer potential associations between microbes and diseases, the proposed HGNNTMDA model adopts an integrated loss strategy. Specifically, Huber loss is utilized as the primary measure to assess the reconstruction error between the model’s predictions and the actual ground truth values. Contrastive Loss operates on the learned feature embeddings, targeting the model’s capacity to discriminate between semantically similar and dissimilar samples. By minimizing intra-class distances and maximizing inter-class distances in the latent space, this loss facilitates more effective separation of Microbe and disease representations—thereby improving the model’s capacity to distinguish biologically relevant associations. By adopting this integrated loss strategy, the model demonstrates superior performance in tasks involving microbial-disease association prediction.

Experimental results and analysis

Experimental setup

The experiments in this research were performed using Python (version 3.11.5, 64-bit) on Spyder (version 5.4.3, conda). The implementations relied on the PyTorch deep learning framework and the DGL graph neural network framework. The computational environment included an Intel(R) Core(TM) i7-4790 CPU @ 3.60GHz with 24GB RAM and an NVIDIA RTX 4070 GPU with 12GB RAM.

Evaluation metrics

To comprehensively assess the predictive performance of the proposed model, multiple evaluation criteria were adopted, including accuracy, specificity, precision, F1-score, the receiver operating characteristic (ROC) curve, and the area under the receiver operator characteristic (ROC) curve (AUC). The detailed computation procedures align with those described by Zhu et al. (2021).

Given the inherent sparsity and imbalance in biological datasets, we implemented a data resampling strategy to artificially balance the training and testing sets, thereby enhancing the robustness of performance evaluation. We employed 5CV to evaluate model accuracy. This validation scheme involves partitioning the dataset into five equally sized folds, where each fold serves as a test set in one iteration, while the remaining four folds are used for training. Such a rotation mechanism guarantees that every sample is utilized for both validation and learning, effectively reducing variance due to data partitioning.

Experimental results on the HMDAD database

In this subsection, we initially establish a set of hyperparameter combinations as the default values for benchmarking (see Table 1). These values are used to perform performance validation experiments on the HGNNTMDA model. Figure 2 illustrates the experiment results for the 5CV experiments on the HMDAD database, utilizing the hyperparameters specified in Table 1.

Table 1 Initial default settings for hyperparameters.

Hyperparameter	Parameter meaning	Parameter value	
lr	learning rate	0.0001	
weight_decay	A regularization technique that adds the L2 norm (sum of squared weights) to the loss function to prevent overfitting.	0.00001	
k_neigs	the number of nearest neighbors to select	13	
clusters	Number of Clusters	9	
n_head	Number of Attention Heads	8	
nlayer	Number of Layers	2	
epochs	Number of Epochs	200	
dropout	Dropout Rate	0.5	

Figure 2 Initial experimental results.

(A) Loss curve. (B) Accuracy curve. (C) ROC curve. (D) Evaluation metric value.

As shown in Fig. 2, the consistent decline in loss alongside the gradual increase in accuracy highlights the HGNNTMDA model’s promising performance on the classification task. The accuracy trend demonstrates a continuous improvement throughout the training process, suggesting that the model progressively enhances its classification ability and becomes more adept at predicting the correct categories. The model achieved an average AUC of 0.9748 across 5CV experiments, which underscores its robustness and generalization ability.

Parameters analysis on HMDAD database

To systematically investigate the sensitivity of the HGNNTMDA model to key hyperparameters, we conducted a comprehensive analysis focusing on several core training hyperparameters, including the number of epochs, learning rate (lr), weight decay, neighborhood size (k_neigs), number of clusters (clusters), number of graph convolution layers (nlayer), and attention heads (n_head). All experiments were implemented using 5CV on the HMDAD dataset to ensure consistency in evaluation. Considering the potential variability introduced by random data partitioning within the 5CV protocol, each experiment was independently repeated 100 times. The reported results represent the mean performance across these iterations, thereby minimizing the impact of sampling fluctuations and enhancing statistical robustness.

First, we conducted a more detailed investigation into the impact of each hyperparameter on the model’s predictive performance. AUC was employed as the primary performance indicator throughout. The exact search intervals and candidate values for each hyperparameter are summarized in Table 2. Figure 3 illustrates the impact of each parameter on model performance.

Table 2 Range of values for hyperparametric analysis.

Hyperparameter name	Values range	
lr	0.0001–0.05	
weight_decay	0.00001–0.005	
k_neigs	[3, 4, 5, 6, 7, 8, 9, 10, 11, 12, 13, 14, 15, 16, 17, 18, 19, 20]	
clusters	[2, 3, 4, 5, 6, 7, 8, 9, 10, 11, 12, 13, 14, 15, 16, 17, 18, 19, 20]	
dropout	[0.2, 0.3, 0.4, 0.5]	
epochs	[100, 200, 300, 400, 500, 600, 700, 800, 900, 1000]	

Figure 3 Hyperparametric analysis.

(A) lr. (B) k_neighs. (C) weight_decay. (D) epoch. (E) dropout. (F) clusters.

The parameter lr determines the step size during parameter updates. Figure 3A shows that an lr of 0.0005 yields the highest AUC. The parameter k_neigs, which specifies the number of neighbors each node is connected to, influences the density of the KNN graph and inter-node relationships. As shown in Fig. 3B, the optimal value for k_neigs is 11. Weight decay, a regularization method that discourages large weights to reduce overfitting, reaches its optimal AUC at 0.00005, as displayed in Fig. 3C. Dropout, another regularization technique that involves randomly omitting neurons during training to combat overfitting, achieves its highest AUC when the training epochs reach 800, as seen in Fig. 3D. Figure 3E shows that a dropout rate of 0.2 produces the highest AUC value. Lastly, the number of clusters, which determines data partitioning and affects graph connectivity, achieves the maximum AUC with 10 clusters, as indicated in Fig. 3F.

In the Transformer encoder, the number of heads (n_head) in the multi-head AM significantly affects model performance. More heads allow the model to compute multiple attention scores in parallel, improving computational efficiency and enabling the capture of diverse representations. However, too many heads can increase computational and memory overhead. The number of layers (nlayer) in the encoder indicates the stacked layers used, impacting the model’s ability to capture complex features and dependencies. While more layers can enhance model representation, they also increase the risk of vanishing or exploding gradients, especially without proper regularization or normalization.

To further examine the joint sensitivity of the HGNNTMDA model, we investigated the interaction between the lr and the number of attention heads (n_head) by adjusting n_head values (2, 4, 8, and 16) alongside a range of lr values, consistent with those listed in Table 2. As illustrated in Fig. 4A, the optimal configuration, yielding the highest AUC, is n_head = 2 and lr = 0.0005. In addition, we assessed the combined impact of lr and the number of layers (nlayer) by varying nlayer from 2 to 8 while exploring different lr values. Figure 4B indicates that the optimal setting is nlayer = 2 and lr = 0.0005. This observation underscores the considerable influence that the interplay between architectural depth and learning rate exerts on the predictive capabilities of the HGNNTMDA model.

Figure 4 Joint parametric experiments.

(A) Joint sensitivity analysis of parameters n_head and lr. (B) Joint sensitivity analysis of parameters nlayer and lr.

After thorough experimentation and critical evaluation of each hyperparameter’s contribution, we chose a set of optimal hyperparameter combinations shown in Table 3. Figure 5 illustrates that fine-tuning the model’s hyperparameters produced a noticeable enhancement in performance. Under 5CV, the average AUC increased by more than 2%. This uplift implies enhanced capacity for generalization and more accurate classification on the HMDAD dataset.

Table 3 Optimal hyperparameter settings for the HMDAD dataset.

Hyperparameter name	Parameter value	
lr	0.0005	
weight_decay	0.00005	
k_neigs	11	
clusters	10	
n_head	2	
nlayer	2	
epochs	800	
dropout	0.2	

Figure 5 Experimental results after parameter optimization.

(A) Loss curve. (B) Accuracy curve. (C) ROC curve. (D) Evaluation metric value.

Experimental results on the disbiome database

In order to fully assess the stability and generalization ability of the proposed model, we conducted five more cross-validation experiments on the Disbiome dataset. Due to the relatively limited size of the HMDAD dataset, in the previous experiments, we performed hyperparameter optimization search by analyzing each important hyperparameter individually. However, the Disbiome database is rich in data, with more complete disease types and microbial types, and the data volume is a bit larger. Therefore, during the experiments, we used the Optuna (Akiba et al., 2019) method to search for optimal hyperparameters within the proposed model. Optuna uses a Bayesian optimization algorithm to help the user find the best combination of hyperparameters in as few experiments as possible by using an intelligent search strategy. The hyperparameter search ranges of Optuna are still using the ranges specified in Table 2 in the previous section. The best hyperparameter combinations obtained by hyperparameter optimization on the Disbiome dataset by the Optuna method are shown in Table 4.

Table 4 Optimal hyperparameter settings for the disbiome dataset.

Hyperparameter name	Parameter value	
lr	0.0002	
weight_decay	0.000006	
k_neigs	3	
clusters	19	
n_head	2	
nlayer	2	
epochs	800	
dropout	0.2	

After setting the model hyperparameters to the optimal hyperparameter combinations, the quintuple cross-validation experiments were repeated several times to evaluate the operational performance of the proposed model on the Disbiome dataset. The corresponding results shown in Fig. 6 indicate that the model performs well on several key evaluation metrics, thus validating the model’s adaptability and predictive ability.

Figure 6 Experimental results on the Disbiome database.

(A) Loss curve. (B) Accuracy curve. (C) ROC curve. (D) Evaluation metric value.

Comparison with other models

In order to evaluate the predictive performs of our proposed HGNNTMDA model, we benchmarked its performance against six well-known methods: KATZHMDA (Chen et al., 2017), NTSHMDA (Luo & Long, 2020), BRWMDA (Yan et al., 2020), NGRHMDA (Huang et al., 2017), GATMDA (Long et al., 2021) and GCNMA (Wang et al., 2023). All comparative experiments were run under the 5CV scheme on both the HMDAD and Disbiome datasets.

Figure 7 presents the results of the comparison. It is worth highlighting that the HGNNTMDA model achieved the highest AUC scores in both datasets. It is demonstrated that the HGNNTMDA model has significant advantages over other competing methods.

Figure 7 AUC of different models on two datasets.

(A) HMDAD, (B) Disbiome.

Ablation experiment

To better understand how the contrastive learning module and the AM each influence the HGNNTMDA architecture, we carried out an ablation study. We conducted 5CV on two datasets: HMDAD and Disbiome. In the ablation experiment, we built two pared-down variants of the original model: (1) HGCN_ATT: retains the AM but does not employ contrastive learning during training;

(2) HGCN_CL: removes attention-related components while preserving the contrastive learning strategy.

By comparing these two variants, we can clearly see how each component contributes to the overall performance of the HGNNTMDA framework. The hyperparameters in the experiments of the two variant models on both datasets still use the best combination of hyperparameters obtained in the previous experiments.

The results of the comparison experiments between the HGNNTMDA model and the two variants of the model are shown in Fig. 8. From the comparative outcomes illustrated in the figure, it becomes apparent that the full HGNNTMDA model delivers consistently higher performance than the two pared-down variant models. Experimental results demonstrate that these components work best in concert and that their joint contribution is essential for achieving robust and stable prediction results.

Figure 8 Comparative performance of HGNNATTCLMDA and its two ablated variants on the HMDAD and Disbiome datasets.

(A–C) Results on HMDAD: (A) HGNN_ATT, (B) HGNN_CL, and (C) HGNNATTCLMDA. (D–F) Results on Disbiome: (D) HGNN_ATT, (E) HGNN_CL, and (F) HGNNAT.

To investigate the impact of various loss function choices on the predictive performance of the model, we designed another set of ablation experiments. We conducted three different experiments using three loss functions: L1 Loss, L2 Loss, and Huber Loss. The results are presented in Table 5.

Table 5 Comparison of performance index values for choosing different loss functions.

	Loss function	AUC	Accuracy	F1_score	Specificity	Precision	
HMDAD	L1 loss	0.5806	0.5000	0.0000	0.5000	0.6667	
L2 loss	0.9724	0.9250	0.9222	0.9234	0.9249	
Huber loss	0.9976	0.9889	0.9890	0.9833	0.9839	
Disbiome	L1 loss	0.5006	0.5000	0.6667	0.0000	0.5000	
L2 loss	0.9332	0.8865	0.8839	0.9092	0.9051	
Huber loss	0.9423	0.8826	0.8815	0.8922	0.8904	
Note:

The best results are marked in bold.

Table 5 presents the mean results derived from 100 independent experimental repetitions. On the HMDAD dataset, the model consistently achieved optimal performance when trained with the Huber Loss function across all runs. In contrast, for the Disbiome dataset, the model utilizing L2 Loss occasionally yielded the best outcomes, although such instances were comparatively infrequent.

Case studies

To further validate the predictive accuracy and reliability of the HGNNTMDA model, we conducted a case study focusing on colon cancer, one of the most dangerous malignant tumors (Chen et al., 2024). The HGNNTMDA model was employed to predict the likelihood of association for each microbial candidate disease. These associations were then ranked in descending order. The top 10 candidate microorganisms for colon cancer were selected for validation and analysis. Figure 9 presents these top 10 candidate microorganisms. All top 10 microorganisms identified by the HGNNTMDA model as associated with colon cancer have been experimentally confirmed, as detailed in Table 6.

Figure 9 Top 10 candidate microorganisms for colon cancer.

Table 6 Predicted top 10 microbes for colon cancer by HGNNTMDA.

Rank	Microbe	Evidence	
1	Helicobacter pylori	PMID: 30430119 (Mansour et al., 2018)	
2	Clostridium difficile	PMID: 21272802 (Kariv et al., 2011)	
3	Staphylococcus aureus	PMID: 34678970 (Ahmad-Mansour et al., 2021)	
4	Staphylococcus	PMID: 36557606 (Wei et al., 2022)	
5	Actinobacteria	PMID: 37633504 (Pongen et al., 2023)	
6	Proteobacteria	PMID: 38329696 (Wang et al., 2025)	
7	Firmicutes	PMID: 33492552 (Chattopadhyay et al., 2021)	
8	Clostridium coccoides	PMID: 29667480 (Gomes, Hoffmann & Mota, 2018)	
9	Stenotrophomonas maltophilia	PMID: 22232370 (Brooke, 2012)	
10	Burkholderia	PMID: 34346791 (Bach et al., 2022)	

Helicobacter pylori and Clostridium difficile have been associated with the initiation and progression of ulcerative colitis, a persistent inflammatory bowel condition that significantly elevates the risk of colorectal malignancy (Kariv et al., 2011; Mansour et al., 2018). In a similar vein, Staphylococcus aureus and other members of the Staphylococcus genus are known to secrete virulence factors that can interfere with host immune regulation and may, under certain pathological contexts, contribute to tumor development (Ahmad-Mansour et al., 2021; Wei et al., 2022). While Actinobacteria have demonstrated therapeutic potential in oncology through the biosynthesis of bioactive metabolites, their relevance to colorectal cancer remains indirect and requires further clarification (Pongen et al., 2023). The phylum Proteobacteria has been found in elevated abundance within dysbiotic intestinal environments and colorectal tumor models, suggesting its participation in inflammation-associated tumorigenesis (Wang et al., 2025). In contrast, a decline in beneficial Firmicutes, especially butyrate-producing taxa, is thought to compromise mucosal barrier integrity, thereby fostering a pro-tumorigenic state (Chattopadhyay et al., 2021). Although Clostridium coccoides has not been directly implicated in colorectal cancer, it is involved in metabolic pathways related to obesity, an established risk factor for cancer development (Gomes, Hoffmann & Mota, 2018). Stenotrophomonas maltophilia, typically classified as an opportunistic pathogen, is frequently isolated from inflamed gastrointestinal tracts and may promote chronic inflammatory conditions conducive to neoplasia (Brooke, 2012). Additionally, Burkholderia species, characterized by diverse metabolic capabilities, are known producers of secondary metabolites that may influence the tumor microenvironment, albeit without direct experimental linkage to colorectal cancer (Bach et al., 2022).

In conclusion, among the microbial candidates predicted by the HGNNTMDA model, at least four demonstrate direct empirical connections to colorectal cancer, while the remaining taxa exhibit indirect or mechanistically plausible associations through inflammatory, immunological, or metabolic pathways. These validation results further demonstrate the robust predictive performance of the HGNNTMDA model and provide valuable insights for microbe-based disease diagnosis and treatment.

Discussion

Over the past few years, researchers have increasingly focused on developing computational strategies to predict associations among microbes, miRNAs, and various human diseases. These algorithmic frameworks have yielded encouraging findings and, to some degree, have propelled progress in biomedical informatics. Still, critical limitations persist. Such as a key concern lies in the scarcity of experimentally verified microbe-disease associations, which results in inherently sparse interaction networks. Because the majority of existing models depend on these known links for training, the performance of such models often deteriorates when applied to sparsely connected datasets.

To mitigate this issue, we present HGNNTMDA, a composite architecture that synergizes hypergraph neural networks, attention-based feature selection, and Transformer encoders. This framework processes Microbe-disease data through a sequence of interlinked modules. The HGNN layer updates node representations by aggregating multi-order neighborhood information. Meanwhile, embedding the Transformer module within the hypergraph topology allows the model to capture non-local semantic relations across heterogeneous biomedical entities. This combination not only boosts predictive reliability but also reduces the model’s sensitivity to noisy or incomplete data.

However, due to the limited number of validated associations, the resulting hypergraph is highly sparse. This sparsity can negatively impact the quality of node embedding, particularly for entities with few connections. To address this challenge, a contrastive learning objective is incorporated under a self-supervised framework. This design allows the model to extract latent structural signals beyond direct supervision, thereby enhancing its generalizability in sparse scenarios. The synergy between attention and contrastive learning boosts the model’s sensitivity to biologically meaningful signals, leading to improved generalization.

We evaluated the predictive performance of HGNNTMDA by comparing it with several established benchmark models. These models include KATZHMDA, NTSHMDA, BRWMDA, NGRHMDA, GATMDA, and GCNMA. Comparative experiments were performed on the HMDAD and Disbiome datasets, using 5CV to ensure robustness. The results show that HGNNTMDA consistently achieves higher predictive accuracy than these benchmark models. Ablation experiments were designed for the HGNNTMDA model, and the results of the ablation experiments on the HGNNTMDA modules showed that removing any of the HGNNTMDA modules resulted in different degrees of degradation of the overall performance of the HGNNTMDA model, which suggests that each of the HGNNTMDA modules designed in this article is effective. Complementary case studies further validated that the predictions made by the model align well with experimentally verified microbe–cancer associations, attesting to its biologically sound nature.

Conclusion

This study introduces a hybrid model HGNNTMDA designed to predict human microbe-disease associations. The model integrates three core components: HGNNs, Transformer-based encoders, and attention-driven feature selection. To assess its effectiveness, we conducted empirical evaluations on the HMDAD and Disbiome datasets. To enhance the robustness of the experimental findings, we implemented a 5CV strategy during model evaluation. The model outperformed six commonly used baseline algorithms in predictive accuracy. Ablation analyses and comparisons with advanced models further proved the approach’s stability and practical value. One important direction for future investigation involves optimizing the structure of the hypergraph neural network and exploring different hyperedge definitions to increase the model’s expressive power.

Supplemental Information

Supplemental Information 1 Experimental data and code for the proposed HGNNTMDA model.

Additional Information and Declarations

Competing Interests

The authors declare that they have no competing interests.

Author Contributions

Rong Zhu conceived and designed the experiments, performed the computation work, authored or reviewed drafts of the article, and approved the final draft.

Yong Wang conceived and designed the experiments, performed the experiments, analyzed the data, performed the computation work, prepared figures and/or tables, authored or reviewed drafts of the article, and approved the final draft.

Junliang Shang analyzed the data, authored or reviewed drafts of the article, and approved the final draft.

Ling-Yun Dai performed the experiments, performed the computation work, prepared figures and/or tables, authored or reviewed drafts of the article, and approved the final draft.

Feng Li performed the experiments, authored or reviewed drafts of the article, and approved the final draft.

Data Availability

The following information was supplied regarding data availability:

The code and raw data are available in the Supplemental File.

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
