# Peer review of "Optimizing transformer-based prediction of human microbe–disease associations through integrated loss strategies"

_PeerJ Computer Science, doi:10.7717/peerj-cs.3098_

## Round 0.1 · original submission · Major Revisions

· Academic Editor

Major Revisions

The reviewers have substantial concerns about this manuscript. The authors should provide point-to-point responses to address all the concerns and provide a revised manuscript with the revised parts being marked in different color.

Reviewer 1 ·

Basic reporting

This paper proposed the HGNNATTCLMDA model for microbial disease prediction, which combines a hypergraph neural network, Transformer, and Huber loss to enhance prediction accuracy.

Strengths:
1. The model design section is clear and well-structured, making it easy to understand the key points of each module.
2. The authors have provided the code for public evaluation, which promotes reproducibility and transparency.

Weaknesses:
1. Unclear research gap: The main challenges in microbial disease prediction are not clearly defined. While the introduction mentions the "complexity and diversity" of microbial and disease data, these challenges are too general. The authors should provide more specific challenges or examples to illustrate these issues.
2. Lack of related work analysis: The paper lists several methods for identifying potential microbial-disease associations but does not analyze them in detail. For instance, how are these methods designed? What unresolved problems remain? A critical evaluation of existing works would strengthen the paper.
3. Abstract formatting: The abstract should summarize the entire paper in one cohesive paragraph rather than dividing it into sections like background, methods, and results.
4. Introduction lacks clarity: The introduction overemphasizes the importance of predicting microbial-disease associations with too many paragraphs, while only briefly mentioning the challenges and the designed model. This makes it difficult to follow. Additionally, the introduction does not clearly explain why the HGNNATTCLMDA model was designed or how it addresses the identified challenges.
5. Rationale for module choice: The HGNNATTCLMDA model appears to be a combination of several existing modules. However, the paper does not adequately justify why these specific modules were chosen or how they contribute to addressing the challenges in microbial-disease prediction.
6. Language issues: The paper contains grammatical errors and uses some words in Chinese. A thorough proofreading and revision are needed to improve language quality and consistency.

Experimental design

See as Basic reporting, besides:
1. Baseline Models: The baseline models compared are outdated (before 2020). The authors should consider including comparisons with newer methods to provide a more relevant and fair evaluation of HGNNATTCLMDA.
2. Dataset: The study evaluates the model using only one dataset (HMDAD). To demonstrate the robustness and generalizability of HGNNATTCLMDA, additional datasets should be included for evaluation. This will strengthen the validity of the findings.

Validity of the findings

See as Basic reporting

Additional comments

NA

Reviewer 2 ·

Basic reporting

1. Since the HGCN from (Gao et al. 2023) is basically a graph neural network it is not accurate to say that it is more adept at managing complex higher-order relationships between nodes than a GNN.

2. The description of the data set is lacking. For instance, what are the feature vectors for microbe or disease?

3. While the PMID is useful in Table 5, you should provide full citations as you are referencing other scientitists work.

4. The acronymn HGNNATTCLMDA is never explicity defined, as it should be before its first use. As a secondary point, the acronymn is so clunky it will likely not get used and if it is used, it will probably be used incorrectly.

5. One Line 150, $H$ is not a set containing all vertices it is the hypergraph. I think you mean $V$ is the set of all vertices.

6. In displayed equations 6, 7, and 8 the choice of notation is uncessarily confusing. For example, $H$ gets used 3 different ways, once as the incidence matrix of the hypergraph (which is an abuse of notation here), once in bold as the incidence matrix again, and once with a superscript indiciating node featuers. There are similar issues with $L$ and $W$.

7. The citation of (Gao et al. 2022) is not an appropriate citation for the idea of a hypergraph. A more common citation for the notion of hypergraphs would be the book of Berge, but other citations could be used.

8. In the description of display (9), $P$ isn't defined and || is not used in the formula, but is defined.

9. In display (5), an $L$ is used instead of $\ldots$.

10. In line 253, you are not introduce the Huber loss as it has already been introduced, you are using it. Also, the citation of (Huang 2022) is not appropriate since you are not referencing anything about the work of Huang.

Experimental design

1. The similarity metric for the constastive learning is not defined.

2. I question the use of the same hyperparameters in the ablation experiment. It seems likely to me that different architectures would need different hyperparameter settings to achieve their best results.

3. Since $K$-means is $NP$-complete to calculate, I assume you used a heuristic methods for finding the partition. The details of that method should be included as well as the values of any hyperparameters used in the method.

4. The constrastive learning approach is not described in sufficient detail to be replicated.

5. The Laplacian proposed and the HCGN approach of Gao are both equivalent to operations on the graph formed from the hypergraph by the weighted cliquen expansion of the graph. In your case, the union of the $K$-means adjacencies and the $K$-nearest neighbors adjacencies -- so not really a hypergraph method.

6. The weight matrix from display (6) is note defined.

Validity of the findings

1. In the conclusion you emphasize that this method is innovative, but you provide little evidence for this in the article. Additionally, you assert that the neural network excels in modelling hihg-dimensional, intricate data but provide no evidence that the data is either high-dimensional or intricate.

---

## Round 0.2 · Major Revisions

· Academic Editor

Major Revisions

There are some remaining major concerns that need to be addressed.

Reviewer 1 ·

Basic reporting

The authors have responded appropriately to all comments and have significantly improved the manuscript. The revised version is well-structured and professionally written, with improved clarity and coherence. Background and related work are now clearly presented with detailed and updated references.

Experimental design

The research question is clearly defined and addresses a relevant gap in microbe-disease association prediction. The revised manuscript enhances methodological transparency and includes more recent baseline comparisons

Validity of the findings

The findings are valid and well-supported by experiments on two datasets. The results are clearly presented, and the added comparisons strengthen the reliability of the proposed method.

Reviewer 2 ·

Basic reporting

Again, the approach being defined is not different from a graph
neural network. This can be seen by rearranging the Laplacian
defined in display (3). $HWD_{e}^{-1}H^T$ is the same as a weighted
matrix, where $u ~ v$ are adjacent if they are in same $K$-means
clustering and the weight is given by the weight of the hyperedge
divided by the size of the hyperedge.

Is HGNNTMDA an acronym for something? If so that is not
defined, if it is simply a collection of letters, a better name
should be chosen.


At several points in the exposition, the same statement is made
2 or even 3 times in a row with slight rewording. The Constructing
Hypergraph section is emblematic of this issue.

Display 2 should be $H_{ik}$.

Display $7$, do you really mean that $sim$ here? So the
numerator is a doubly exponential term involving the temperature
twice?

On line 303 it is not clear what $[0,1][0,1][0,1]$ is
supposed to mean, do you mean $[0,1]^3$.

Experimental design

A quick search of the PyTorch and DGL documentation did not find the function construct_kmeans_graph. If the parameters for the kmeans clustering were fit with Optuna or other methods, this should be specified and the heuristic used be described.

Validity of the findings

The validity of the findings would be enhanced by some discussion association probability for colon cancer and the actual association probability. The citations provided do not seem (for the most part) have studied the association with colon cancer of these microbiomes.

---

## Round 0.3 · Minor Revisions

· Academic Editor

Minor Revisions

There are some minor concerns that need to be addressed.

Reviewer 2 ·

Basic reporting

I still disagree on calling this a hypergraph method, but it is clear it is a philosphical difference. I suspect that a certain audience will dismiss this work out of hand because of the terminology, but that is a choice the author's have made.

Given the audience, it would be better to not use miRNA and rRNA and instead use the full words (i.e., messenger RNA instead of mRNA). I don't know what the ``mi'' or ``r'' stand for so I can't assess whether these are typos or not.

The grammar of the last sentence before the Materials and Methods section needs to be checked.

On line 190, Hypergraph should be plural in this context.

On lines 221--235 you refer to contrast learning and contrastive learning without a clear pattern for when you use one form and when you use the other. You should check that you are using the form you want to use in all cases. I believe that changing them all to constrative learning would be appropriate.

On line 303-304, the sentence starting ``The commonly used...'' is a fragment.

Experimental design

No comment

Validity of the findings

No comment

---

## Round 0.4 · accepted · Accept

· Academic Editor

Accept

The authors have addressed all the concerns, and I recommend accepting this manuscript.